# Optimal Lead Position in Patch-Type Monitoring Sensors for Reconstructing 12-Lead ECG Signals with Universal Transformation Coefficient

**DOI:** 10.3390/s20040963

**Published:** 2020-02-11

**Authors:** Dongseok Lee, Hyunbin Kwon, Hongji Lee, Chulhun Seo, Kwangsuk Park

**Affiliations:** 1Interdisciplinary Program in Bioengineering, Seoul National University, Seoul 03080, Korea; azuremoon@bmsil.snu.ac.kr (D.L.); chasekwon@bmsil.snu.ac.kr (H.K.); 2Institute of Medical and Biological Engineering, Medical Research Center, Seoul National University, Seoul 03080, Korea; 3Mobile Communication Business, Samsung Electronics Co., Ltd., Suwon 16677, Korea; hongjidan@bmsil.snu.ac.kr; 4School of Electronic Engineering, Soongsil University, Seoul 06978, Korea; chulhun@ssu.ac.kr; 5Department of Biomedical Engineering, College of Medicine, Seoul National University, Seoul 03080, Korea

**Keywords:** reconstructed electrocardiogram, 12 lead ECG, wearable patch device, artificial neural network, universal coefficient

## Abstract

The aim of this study was to reconstruct a 12-lead electrocardiograph (ECG) with a universal transformation coefficient and find the appropriate electrode position and shape for designing a patch-type ECG sensor. A 35-channel ECG monitoring system was developed, and 14 subjects were recruited for the experiment. A feedforward neural network with one hidden layer was applied to train the transformation coefficient. Three electrode shapes (5 cm × 5 cm square, 10 cm × 10 cm square, and right-angled triangle) were considered for the patch-type ECG sensor. The mean correlation coefficient (CC) and minimum CC methods were applied to evaluate the reconstruction performance. The average CCs between the standard 12-lead ECG and reconstructed 12-lead ECG were 0.860, 0.893, and 0.893 for a 5 cm × 5 cm square, 10 cm × 10 cm square, and right-angled triangle shape. The right-angled triangle showed the highest performance among the considered shapes. The results also suggested that the bottom of the central area of the chest was the most suitable position for attaching the patch-type ECG sensor.

## 1. Introduction

An electrocardiograph (ECG) is used to measure irregular rhythms during cardiovascular activity and diagnose heart diseases. In particular, a 12-lead ECG is utilized as a gold standard tool to diagnose cardiovascular diseases such as myocardial infarction and atrial fibrillation.

A 12-lead ECG comprises ten electrodes: three electrodes are attached on the patient’s left arm, right arm, and left leg to measure the limb leads (lead I, II, and III) and augmented limb leads (aVR, aVL, and aVF), six electrodes are placed on the patient’s chest to calculate the precordial lead (V1, V2, …, V6), and one electrode acts as the ground reference [1]. Information regarding the heart activity can be obtained using a 12-lead ECG. 

However, the correct use of a 12-lead ECG requires special medical knowledge because the electrodes should be attached at the exact position. In addition, multiple electrodes on the patient’s limb and chest may disturb the patient’s free movement and make the patient feel uncomfortable during long-term monitoring. 

Several studies have suggested various methods to overcome these issues. A standard 12-lead ECG can be reconstructed by using a reduced number of leads with transformation matrices. One of the methods consists of using the subsets of a conventional 12-lead ECG [2,3,4]. Nelwan et al. used a multiple linear regression (MLR) with a reduced number of leads to reconstruct a 12-lead ECG [4]. The correlation coefficient (CC) was calculated between the reconstructed ECG and the reference ECG by changing the number of precordial leads. The most suitable leads containing four electrodes were the lead I, II, and V2, and the CC was 0.912. Lee et al. used the state-space model to reconstruct the precordial lead ECG from limb leads [5]. The average CC from V1 to V6 was 0.90. Wang et al. used the convolutional neural network model to derive standard 12-lead ECG from 3-lead ECG [6]. The average CC of precordial leads was 0.949. However, the use of subsets of the 12-lead ECG requires multiple electrodes to ensure acceptable reconstruction performance. Therefore, it may be bulky as the electrodes need to be attached on the limb or at the correct position on the chest.

To overcome these limitations, other studies have used special leads such as vectorcardiography (VCG) and EASI systems. The Frank VCG system is based on an orthogonal lead system. This system requires seven electrodes, using a linear transformation matrix [7]. Willems reported that the diagnostic accuracies of ECG and VCG were 80.3% and 79.3%, respectively, according to a five-group logistic classification [8]. However, the application of the VCG lead system is complicated for non-expert. The EASI system is more practical and easier to use than the VCG system. The EASI system uses the E, A, and I electrode positions of the Frank lead system [9], and S electrode position on the upper end of the sternum to derive 12-lead ECG [10,11,12]. This system is less sensitive to the motion artifact, and easier to attach electrodes than the standard 12-lead ECG system. However, the VCG and EASI systems require the electrodes to be attached at the correct location. 

Recently, other studies have reported new systems to overcome the disadvantages of the above systems. Finlay et al. proposed a new method for reconstructing a 12-lead ECG from an eigenvector by using a principal component analysis [13]. The 12-lead ECG was reconstructed with three ECG vectors selected from the body surface potential map, and the median CC was 0.907. Trobec et al. suggested a 12-lead ECG reconstruction algorithm with differential leads [14]. The 12-lead ECG was synthesized with the MLR from the electrodes attached to the patient’s chest including the back of the subject. Hadzievski et al. developed a mobile device with five electrodes so that the patients could attach the device on their chest [15]. Three electrodes were placed on the side attached to the chest near the sternum, and two electrodes were placed on the front side. The 12-lead ECG was reconstructed with three leads that were generated when the patient touched the electrode with both index fingers. The results showed that 80.2% of the reconstructed ECGs were similar to the reference 12-lead ECG, with no differences noticed from observers. Dos Reis et al. proposed a device that reconstruct the 12-lead ECG using a MR-compatible ECG sensor network [16]. The study showed that the 12-lead ECG can be reconstructed with 4 MR-compatible sensors inside MR with the mean correlation coefficient of 0.887. However, this method is not suitable for daily or long-term monitoring because it requires the subject’s cooperation to measure the ECG signals. Some of the previous 12-lead ECG reconstruction methods are summarized in Table 1.

In addition, technological advances have led to the development of portable devices for daily ECG monitoring such as the Holter monitor. Most of these portable devices are based on a patch-type sensor [17,18,19,20], which can easily be used in daily situations. Bifulco et al. proposed a portable device for continuous monitoring of ECG signals and patient motion during daily life with Bluetooth communication [21]. However, this device was designed to collect ECG signals from only one lead and measure only the QRS complex of the ECG signal or obtain only the heart rate information. In addition, the robustness of position should be considered because the patch-type sensor may be misplaced by non-expert.

The possibility of reconstructing a 12-lead ECG by using a single patch-type device was investigated in our previous study [22]. A 35-channel ECG monitoring system was developed and an artificial neural network (ANN) was utilized to reconstruct the 12-lead ECG. The ANN model was trained to generate a personalized transformation coefficient.

The accuracy of the standard 12-lead ECG reconstruction can be enhanced by using personalized coefficients as the individual electrical characteristics, location of the source dipole, and shape of the volume conductor are different for each individual [23,24]. However, calculating individual coefficients requires personal calibration with a simultaneous monitoring of the standard 12-lead ECG. The calibration process is required before the first measurement, or even for each measurement if the patient’s body shape changes. This method is time-consuming and cannot be used in emergency situations. 

On the other hand, the reconstruction performance is lower when the universal transformation matrix is applied. However, once the universal transformation matrix is calculated, it can be used for any subject without simultaneous 12-lead ECG calibration. The universal 12-lead ECG reconstruction method can be easily used in daily life or during emergencies.

Tomasic et al. have investigated universal positions of electrodes for reconstructing 12-lead ECG signal [23]. However, the study did not consider the shape of the electrode, which requires a bulky system and may be uncomfortable for the patient. Moreover, the possibility should be considered that the patient may not attach the electrodes in the right position when designing patch type device. In this study, we investigated a patch-type electrode model that can be easily applied to reconstruct standard 12-lead ECG in daily life or during emergencies. The appropriate shape for patch-type ECG sensor was considered, and both linear and non-linear models were compared based on the universal transformation coefficient and the universal electrode position. In addition, the robustness of position was considered in case of the misplacement of the patch-type device.

## 2. Materials and Methods

### 2.1. Participants

Fourteen healthy male subjects were recruited for the experiment. Detailed information about the subjects is outlined in Table 2. No medical records or heart-related diseases were reported. Along with the Declaration of Helsinki, a written informed consent was obtained from each subject before the experiment. This study was approved by the Institutional Review Board of Seoul National University Hospital (IRB No.1510-047-710).

### 2.2. Experimental Protocol

An ECG monitoring device that can measure 35-channel chest ECG signals was developed with a commercial ECG sensor module (PSL-iECG, PhysioLab, Korea). The PSL-iECG module is composed of a high-pass filter (cutoff frequency of 0.3 Hz), a low-pass filter (cutoff frequency of 35 Hz), a notch filter (cutoff frequency of 60 Hz), and gain of 500 *V*/*V*. The module was powered by a battery with a DC-DC converter. The data was collected by a data acquisition board (NI-DAQ 6255, National Instruments, Austin, TX, USA). More details are described in our previous paper [22]. The conventional 12-lead ECG was also measured with the same device.

Thirty-five Ag/AgCl electrodes were attached to each subject’s chest. The arrangement of electrodes is described in Figure 1. The 16th electrode was placed at the intersection of the subject’s clavicle and sternum. In addition, three electrodes for the limb leads (lead I, II, and III) and six electrodes for the precordial leads (V1, V2, …, V6) were placed at the appropriate positions through guideline. The ground electrode was attached on the right leg of the subject. Previous studies reported that the minimum inter-electrode distance to obtain a reliable and strong ECG signal should be at least 5 cm [25]. Therefore, the inter-electrode distance was set to 5 cm. 

The standard 12-lead ECG signal was collected simultaneously as the reference. In certain situations, the locations of the electrode for obtaining the reference ECG signals and those of the proposed ECG monitoring system overlapped. In such cases, both the reference ECG and chest ECG signal were acquired by using the same electrodes. The leads from V1 to V6 were referenced to Wilson’s central terminal (WCT) that was generated by lead I, II, and III as follows:(1)$$\mathrm{WCT}=\frac{\mathrm{LA}+\mathrm{RA}+\mathrm{LL}}{3}$$
(LA: left arm; RA: right arm; LL: left leg)

After attaching the electrodes, the subjects were asked to sit comfortably on an armchair and the signals were collected twice for 2 min when the subjects were at rest. All the signals were collected with a data acquisition board (NI-DAQ 6255, National Instruments, USA) with a sampling rate of 250 Hz.

### 2.3. Data Processing

#### 2.3.1. Data Preparation

A second order butterworth bandpass filter with a 0.5–35 Hz cutoff frequency was applied to each signal to remove the baseline wandering and motion artifacts. A 20-s-long segment was extracted from each measurement of each subject. The start point of the segment was randomly selected. The segments of all 14 subjects were merged into a 280 s signal to train the reconstruction model and calculate the universal transformation coefficients. The merged signal from the first measurement was used as the training set of the reconstruction model, and the merged signal from the second measurement was used to test the performance of the model. 

The augmented leads (aVR, aVL, and aVF) were calculated by using three limb leads (lead I, II, and III) as follows [1]:(2)$$\mathrm{aVR}=-\frac{\mathrm{lead}\;I+\mathrm{lead}\;\mathrm{II}}{2}$$
(3)$$\mathrm{aVL}=\frac{\mathrm{lead}\;I-\mathrm{lead}\;\mathrm{III}}{2}$$
(4)$$\mathrm{aVF}=\frac{\mathrm{lead}\;\mathrm{II}+\mathrm{lead}\;\mathrm{III}}{2}$$

#### 2.3.2. Chest Leads

The detailed method for generating a chest lead (CL) is illustrated in Figure 2. Four of the 35 electrodes were selected and the signals from the electrodes were subtracted to calculate the CLs. Three independent CLs were generated from four electrodes. The total number of possible combinations for selecting four electrodes among 35 electrodes was $\left(\begin{array}{c} 35\\ 4 \end{array}\right)$ = 52,360. The 12-lead ECG was reconstructed by using all the 52,360 combinations, and the performance of all the combinations was evaluated.

Further investigation was performed by using a limited number of combinations to find the appropriate position and shape of the electrodes and form a patch-type ECG sensor. 

The following shapes were determined as appropriate for the ECG patch:A 5 cm × 5 cm square (e.g., electrode combination (1, 2, 6, and 7)). The number of combinations for this shape was 6 × 4 = 24.A broader 10 cm × 10 cm square (e.g., electrode combination (1, 3, 11, and 13)): The number of combinations for this shape was 5 × 3 = 15.A right-angled triangle shape in a 10 cm × 10 cm square area (e.g., electrode combination (1,7,11, and 13): Four orientations were considered. The right-angled triangle shape has four orientations, so the number of combinations for this shape was 5 × 3 × 4 = 60.

### 2.4. ECG Reconstruction Model

An MLR and ANN were applied to reconstruct the standard 12-lead ECG from the CLs. The same size of input data was applied to each method to train each model. The independent CLs were normalized by using a zero mean and unit standard deviation before modelling.

#### 2.4.1. MLR Model

MLR is commonly used to model a linear system with the relationship between one dependent variable and one or more independent variables. An MLR model can be expressed as follows:(5)y=bX+ε
where *y* is the dependent variable, *X* is an independent variable matrix with a constant value for biasing, *b* is the conversion matrix, and *ε* is the error of each observation. The conversion matrix b is calculated with the least-square method.

#### 2.4.2. ANN Model

An ANN is a nonlinear method used to model variables. An ANN is known as a useful model to define nonlinear problems and generalized the problem than linear methods. As the body composition is different for each individual, the ANN model was expected to describe the universal 12-lead ECG conversion matrix more precisely.

A feedforward neural network has one input layer, one or more hidden layer, and one output layer. The ANN model employed in this study has three neurons for each independent CL in the input layer and 12 neurons of each 12-lead ECG in the output layer. The optimal number of neurons in the hidden layer was investigated, and six neurons were determined empirically. A logarithmic sigmoid activation function was chosen for the hidden layer, and a softmax function was chosen for the output layer. The weights and bias of the neuron were initialized by using random values at the beginning of the training process and updated with the back-propagation method.

As the weights and biases of the neurons in the hidden layer were initialized by using a random value, the model was trained to the different local minima each time. The model was trained five times with different initial random values, and the outputs of the five models were ensemble-averaged to generate universal output.

### 2.5. Evaluation Metrics

Following the training process, the MLR and ANN models were applied to the test set to reconstruct the standard 12-lead ECG. The CC and root mean square error (RMSE) were calculated between the reference 12-lead ECG signal and the reconstructed signal to compare two signals. The mean and median values of the CC were calculated between the segment of the reconstructed ECG and the segment of reference ECG at the same time for each lead and each subject. To analyze the performance during the reconstruction of all the 12-lead ECG signals, the electrode combination that showed the highest minimum CC (minCC approach) was employed along with the electrode combination showing the highest mean CC (meanCC approach). Also, the R-square value and intraclass correlation coefficient (ICC) was calculated to evaluate the performance of the reconstruction model.

Additionally, a statistical analysis was performed by a means of Wilcoxon’s signed rank test. The differences of the CCs and RMSEs between each shape was observed to compare the performance to reconstruct 12-lead ECG.

## 3. Results

Figure 3 presents the electrode combination that showed highest CC for each shape. As observed in all electrode combinations, the highest mean CC was acquired when the length of the CL was large, and the electrodes were spread over the chest. The minimum CC was the highest when the electrodes were located at the wide line starting from the left clavicle to the right abdomen in both the ANN and MLR models.

The electrode combination that showed the highest CC in the 5 cm × 5 cm square area is described in Figure 3b. The electrode combination located on the upper left chest showed the highest mean CC in both the ANN and MLR models. However, the minimum CC was the highest when the electrode was located on the lower part, at the center of the chest. 

In the 10 cm × 10 cm area, the electrode position was similar to that in the 5 cm × 5 cm square area in terms of the mean CC for the ANN model. However, in case of the minimum CC, the CC was the highest when the electrode position was at the center right part of the chest in both the ANN and MLR models.

In the triangular area, even though there was a difference in terms of the direction between the ANN and MLR for the mean CC, the correlation was the highest when the location of the electrodes was on the left side of the chest. However, in case of the minimum CC, the shape was the same in both ANN and MLR, and the CC was the highest when the electrodes were located at the center of the chest.

Table 3 presents the reconstruction performance of the electrode combination and shows the highest mean and minimum CCs for each shape. The highest mean CC was 0.954 among all combinations in the ANN. The reconstruction performance was higher when the electrode combination was in the 10 cm × 10 cm area than in the 5 cm × 5 cm area, and the highest mean CC value among the 10 cm × 10 cm square combinations was 0.864. Even though the triangle had a smaller area than the 10 cm × 10 cm square shape, the CC for the triangular shape was higher than that for the square shape.

The ANN model showed better reconstruction performance than the MLR model for all shapes. The differences in the CCs of each model were small when the area of the electrode combination was large. The ANN model showed much higher performance compared to the MLR model when the area was small. Among the shapes considered to design a patch-type ECG sensor, the triangular shape showed the highest mean CC and minimum CC.

The RMSE of the ANN model was lower than the that of the MLR model. The lowest RMSE value of all the combinations was 49.3 μV and 52.3 μV in the meanCC approach and the minCC approach, respectively. Although the CC was higher in the right-angled triangle shape than the 10 cm × 10 cm square shape, the RMSE was higher.

As observed in Figure 4, the difference between the 5 cm × 5 cm square and 10 cm × 10 cm square shapes and the difference between the 5 cm × 5 cm square and right-angled triangular shapes were statistically significant in terms of CC and RMSE in the meanCC method. Moreover, the CCs presented the same tendency in minCC methods. However, the difference between the RMSE of the 10 cm × 10 cm square shape and that of the triangular shape was statistically significant.

Also, the Bland-Altman plot of the reconstructed 12-lead ECG signal from the model that showed highest CC in minCC approach is illustrated in Figure 5. The mean difference (bias) and the linearity of the difference were not significant for all three model, and the acceptable error limit was ±0.14 mV, ±0.13 mV, and ±0.13 mV for 5 cm × 5 cm square, 10 cm × 10 cm square, and right-angled triangular shape, respectively. 

## 4. Discussion

### 4.1. Reconstruction Quality of Each Lead

The reconstruction quality of each 12-lead signal at the location that showed highest CC is shown in Figure 6. The overall reconstruction quality was relatively higher with the meanCC method than with the minCC method, especially for precordial leads (V1, …, V6). It means that the reconstruction performance was high when the electrodes were located on the left side of the chest, near the heart of the patient. However, the CCs of lead III and augmented leads were relatively poor at that location. The CC, RMSE, and R-square values of each 12-lead ECG were shown in Table 4 and Table 5. The mean CCs of a reconstructed aVL signal were 0.65, 0.66, and 0.71 for the 5 cm × 5 cm square, 10 cm × 10 cm square, and right-angle triangle, respectively. The reconstruction performance in aVL signal reconstruction was poor while the mean CC of other augmented leads (aVR and aVF) was higher than 0.86. This is because the reconstruction performance of leads I and III was poor with the meanCC method. 

Concerning the minCC method, the electrode position that showed the highest minimum CC was the center or left side of the chest. At these locations, the CCs of lead III and augmented leads were higher, and the average CC was 0.89 in the 10 cm × 10 cm square and right-angle triangular shapes. Although the RMSE was lower in the 10 cm × 10 cm shape, the minimum CC was 0.82 in the lead aVL for the electrode combination (7, 9, 17, 19) while the minimum CC was 0.85 for the electrode combination (12, 14, 18, 22).

### 4.2. Robustness of the Position

It is important to consider the robustness of the electrode position for quick use in emergencies or for non-expert use because the electrodes may be attached at a slightly different position. In Figure 7, the reconstruction quality considering the errors in the positioning of the electrodes was calculated by averaging the results when the electrode positions were shifted with a distance of one electrode in the up, down, left, and right directions. As observed in Figure 7a, the mean CC was high when the electrodes were located on the left side of the chest or bottom part of the center of the chest. The highest mean CC was 0.894. However, the minimum CC was low for the left side of the chest, as observed in Figure 7b. The minimum CC value for this location was 0.702. The highest value of the minimum CC was acquired when the electrodes were located on the bottom of the center of the chest, and its value was 0.757. 

The reconstruction performance of each 12-lead ECG signal is shown in Figure 7c. When the electrodes were placed on the left side of the chest, the reconstructed lead I and precordial leads showed a high correlation. However, lead III showed a low CC, which causes poor reconstruction quality in aVL and aVF when the electrodes were attached on the left side or upper side. Otherwise, the performance of all the leads were similar when the electrodes were placed on the bottom of the center of the chest.

For the bottom of the center of the chest position (electrode combination (13, 15, 23, 25)), the analysis of variance (ANOVA) was conducted to examine the difference of CC when the electrode positions were shifted with a distance of one electrode in the up, left, and right directions compared to the bottom of the center of the chest position. The difference between the orientations is summarized in Table 6. When the electrode combination was (13, 15, 19, 23), the difference of CC between the leads was not significant. Otherwise, there was at least one lead for which the difference of CC was significant for other orientations.

### 4.3. Comparison to Previous Reconstruction Algorithms

The comparisons with previous results are summarized in Table 7. Nelwan et al. used a reduced lead set to the synthesized standard 12-lead ECG [4]. The median CC was 0.912 with three leads (lead I, II, and V2) with four electrodes. Finlay found eigenleads with body surface potential map (BSPM) by using a principal component analysis to reconstruct the 12-lead ECG, and the median CC was 0.907 [13]. Those methods were based on universal transformation matrices. However, the leads are unsuitable for the patch type device because they were located far away from each other. Tomašić examined the universal electrode position by using a personalized transformation matrix [23]. The minimum median CC was 0.84 for the universal position. However, the result is inappropriate for the patch type ECG sensor because the distance between the electrodes was different depending on the body size of each subject and the length of selected leads were too long.

## 5. Conclusions

Herein, a 12-lead ECG reconstruction method was developed with the universal transformation matrix. Both linear and nonlinear methods were adopted for training the reconstruction model. The nonlinear model showed higher performance for reconstructing 12-lead ECG signals. In addition, the lead set was applied on the chest to optimize the shape and location of a patch-type ECG sensor.

Electrode combinations with three shapes were considered for designing the patch-type ECG sensor. The right-angled triangle showed the highest performance among the considered shapes. The median CC among all the 12 leads was 0.920, but the reconstruction performance in the lead III and aVL was poor. The minimum CC was slightly lower, but the CC for lead III and aVL showed considerable improvement. In addition, the results suggested that the bottom part of the center of the chest was the most suitable position for attaching the patch-type ECG sensor considering the robustness of the positioning electrodes. 

The reconstruction algorithm described in this study was conducted with data from healthy subjects only. It is necessary to apply this method to patients with cardiovascular problems to confirm that the 12-lead ECG signals including arrhythmic events were reconstructed correctly. The transformation coefficients could be more informative if the data of patients and female subjects were also included. Patients with cardiovascular problems will be included in future work.

The high-pass filter with a 0.05 Hz cutoff frequency is recommended for ST segment analysis according to the guidelines for processing ECG digital data by American Heart Association (AHA) [26]. The device that was used to measure the ECG signals has a high-pass filter with a 0.3 Hz cutoff frequency to remove the baseline wandering, which may influence the ST segment. Although the high-pass filter used in this study has a 0.5 Hz cutoff frequency, the phase distortion was canceled because the filtering algorithm in MATLAB software is a forward-backward filter. Also, the guideline of AHA recommended 500 Hz for the sampling rate of ECG signal. However, Baumert et al. found that the error in the ECG signal measured with sampling rates of more than 200 Hz was statistically non-significant [27]. Though our method is not suitable for diagnostic ECG because of these limitations, the aim of this study was to examine the possibility of the universal electrode position and shape for 12-lead ECG reconstruction, not to make a device for diagnose cardiovascular diseases. This is a first step for the verification of the system, and the measurement device will be improved by using a lower cutoff frequency such as 0.05 Hz, and other methods will be applied to remove baseline wandering in the further study.

## Figures and Tables

**Figure 1 sensors-20-00963-f001:**
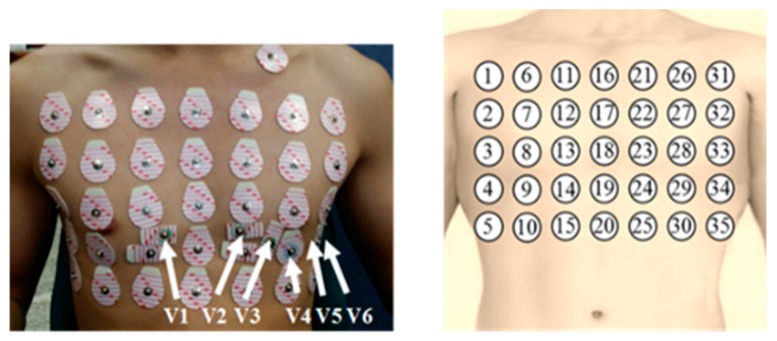
Locations and numberings of the electrodes on the subject’s chest.

**Figure 2 sensors-20-00963-f002:**
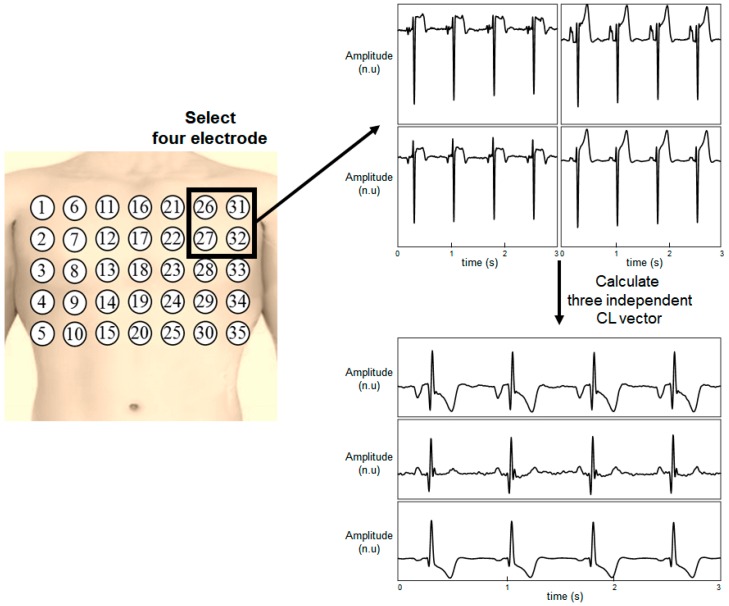
Calculation of the chest leads from four electrodes.

**Figure 3 sensors-20-00963-f003:**
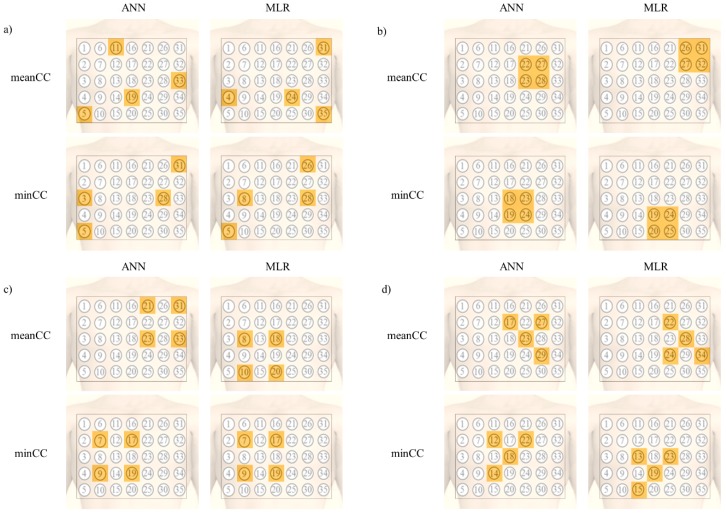
Electrode combination that showed the highest CC for each shape; (**a**) all electrode combinations, (**b**) 5 cm × 5 cm square shape combination, (**c**) 10 cm × 10 cm square shape combinations, and (**d**) right-angled triangular shape combinations.

**Figure 4 sensors-20-00963-f004:**
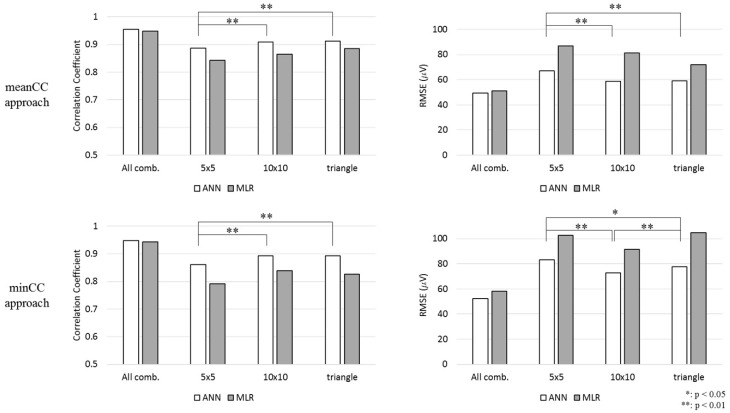
CC and root mean square error (RMSE) values of the electrode combination that showed the highest CC for each shape.

**Figure 5 sensors-20-00963-f005:**
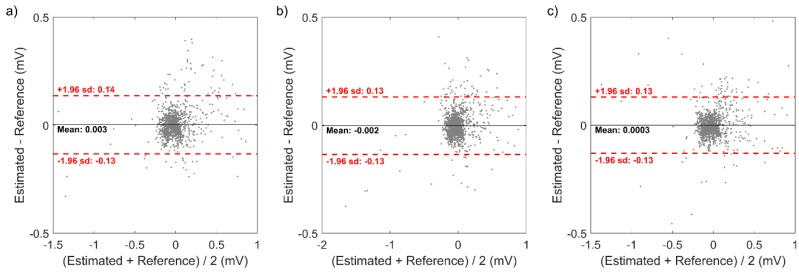
Bland-Altman plot of the reconstructed 12-lead ECG signal from the model that showed highest CC in minCC approach: (**a**) 5 × 5 square, (**b**) 10 × 10 square, and (**c**) right-angled triangular shape.

**Figure 6 sensors-20-00963-f006:**
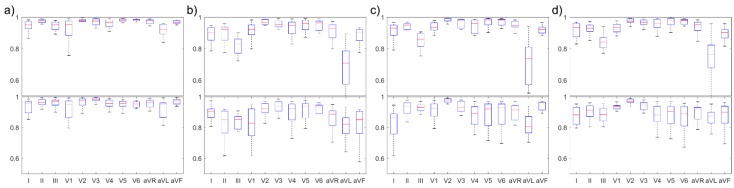
CCs of each reconstructed 12-lead ECG signal at the location that showed the highest CC: (**a**) all combinations, (**b**) 5 × 5 square, (**c**) 10 × 10 square, and (**d**) right-angled triangle. The red line indicates the median value. (upper column: CCmean approach, lower column: CCmin approach).

**Figure 7 sensors-20-00963-f007:**
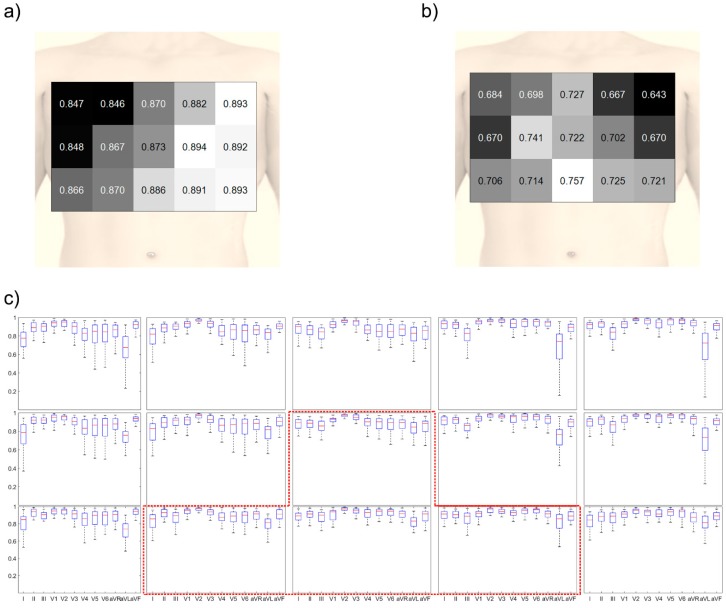
Reconstruction quality when considering the robustness of position: (**a**) colored map obtained by using the mean CC value, (**b**) colored map obtained by using the min CC value, and (**c**) reconstruction quality of the 60 triangular shapes according to the electrode position. The red-dotted line indicates the suggested position.

**Table 1 sensors-20-00963-t001:** Some of the previous 12-lead ECG reconstruction methods.

Study	N	Method	CC
Nelwan [4]	234	Reduced lead sets	0.912 (median)
Lee [5]	290	Reduced lead sets	0.900 (mean)
Finlay [13]	744	Eigenleads	0.907 (median)
Trobec [14]	30	Differntial leads	0.979 (median)
Hadzievski [15]	192	Transtelephonic system	-

**Table 2 sensors-20-00963-t002:** Summary of the subjects’ characteristics (mean ± standard deviation).

Age (Year)	Chest (cm)	Mean Heart Rate (beat/min)
27.4 ± 3.9	93.0 ± 6.5	72.9 ± 8.2

**Table 3 sensors-20-00963-t003:** Reconstruction performance of the electrode combination that showed the highest correlation coefficient (CC).

Approach	Shape	MLR					ANN				
		Mean CC (SD)	Median CC	RMSE (μV)	Mean R^2^	Mean ICC	Mean CC (SD)	Median CC	RMSE (μV)	Mean R^2^	Mean ICC
**meanCC approach**	All comb.	0.948	0.967	51.1	0.87	0.93	0.954	0.970	49.3	0.91	0.95
		(0.07)					(0.05)				
	5 × 5	0.842	0.915	86.8	0.65	0.77	0.887	0.924	67.1	0.78	0.85
		(0.23)					(0.13)				
	10 × 10	0.864	0.916	81.2	0.70	0.81	0.909	0.946	58.5	0.82	0.89
		(0.17)					(0.13)				
	triangle	0.886	0.936	72.0	0.75	0.85	0.912	0.942	58.9	0.83	0.90
		(0.14)					(0.11)				
**minCC approach**	All comb.	0.943	0.958	58.1	0.84	0.91	0.947	0.961	52.3	0.90	0.95
		(0.06)					(0.06)				
	5 × 5	0.792	0.858	103.0	0.58	0.72	0.860	0.892	83.2	0.70	0.84
		(0.19)					(0.10)				
	10 × 10	0.839	0.874	91.7	0.60	0.74	0.893	0.923	72.7	0.75	0.85
		(0.14)					(0.09)				
	triangle	0.827	0.894	105.0	0.66	0.79	0.893	0.920	77.6	0.79	0.89
		(0.18)					(0.08)				

**Table 4 sensors-20-00963-t004:** CC, RMSE and R-square values of each 12-lead ECG for the meanCC approach with the ANN model.

		I	II	III	V1	V2	V3	V4	V5	V6	aVR	aVL	aVF	Mean	SD
**Mean CC**	5 × 5	0.89	0.90	0.82	0.92	0.96	0.95	0.93	0.94	0.94	0.91	0.65	0.83	**0.89**	0.08
	10 × 10	0.91	0.93	0.84	0.93	0.98	0.97	0.95	0.96	0.96	0.95	0.66	0.87	**0.91**	0.08
	Tri.	0.92	0.92	0.85	0.93	0.97	0.96	0.95	0.96	0.96	0.94	0.71	0.86	**0.91**	0.07
**RMSE (μV)**	5 × 5	49.4	73.6	71.1	64.6	84.3	79.6	88.3	65.9	56.7	52.6	51.7	66.9	**67.1**	12.4
	10 × 10	43.4	63.1	68.4	56.5	63.6	71.4	83.1	56.2	46.4	41.7	47.5	61.1	**58.5**	11.9
	Tri.	41.6	63.2	64.4	61.1	73.4	73.4	75.2	57.4	48.4	42.2	45.9	60.4	**58.9**	11.5
**R^2^**	5 × 5	0.81	0.76	0.45	0.76	0.94	0.92	0.88	0.92	0.89	0.84	0.56	0.62	**0.78**	0.15
	10 × 10	0.87	0.83	0.47	0.83	0.97	0.95	0.91	0.93	0.93	0.91	0.58	0.68	**0.82**	0.15
	Tri.	0.87	0.84	0.53	0.79	0.95	0.94	0.93	0.95	0.94	0.91	0.64	0.71	**0.83**	0.13

**Table 5 sensors-20-00963-t005:** CC, RMSE and R-square values of each 12-lead ECG for the minCC approach with the ANN model.

		I	II	III	V1	V2	V3	V4	V5	V6	aVR	aVL	aVF	mean	SD
**Mean CC**	5 × 5	0.89	0.83	0.82	0.81	0.89	0.92	0.90	0.91	0.89	0.86	0.81	0.79	**0.86**	0.04
	10 × 10	0.83	0.93	0.89	0.89	0.97	0.93	0.87	0.88	0.89	0.90	0.82	0.90	**0.89**	0.04
	Tri.	0.88	0.90	0.87	0.92	0.97	0.94	0.88	0.88	0.87	0.90	0.85	0.86	**0.89**	0.03
**RMSE (μV)**	5 × 5	45.5	89.9	69.9	84.2	134.7	108.6	114.5	92.7	78.6	62.2	41.1	76.3	**83.2**	26.3
	10 × 10	57.6	61.1	48.7	65.4	75.8	106.0	128.3	102.0	82.4	54.7	42.8	47.0	**72.7**	25.9
	Tri.	46.7	74.8	62.0	64.9	92.5	106.7	128.3	108.2	89.3	54.4	38.5	64.7	**77.6**	26.3
**R^2^**	5 × 5	0.77	0.65	0.51	0.59	0.81	0.86	0.82	0.78	0.74	0.73	0.63	0.56	**0.70**	0.11
	10 × 10	0.64	0.83	0.72	0.71	0.93	0.84	0.73	0.71	0.70	0.76	0.54	0.84	**0.75**	0.10
	Tri.	0.84	0.80	0.63	0.79	0.93	0.87	0.78	0.80	0.78	0.84	0.70	0.74	**0.79**	0.07

**Table 6 sensors-20-00963-t006:** *P*-values of the ANOVA analysis of the CCs for different orientations of the triangular shape.

Electrode Combination	*p* < 0.05	*p* < 0.01
(13, 15, 19, 23)	-	-
(13, 19, 23, 25)	V6	-
(13, 15, 19, 25)	II	-
(15, 19, 23, 25)	V4, V6	V5

**Table 7 sensors-20-00963-t007:** Comparisons with previous results.

Study	Subjects	Median CC (Interquartile Range)	Method	Algorithm
Nelwan [4]	N=234 (patients)	0.912 (0.858, 0.950)	MLR	Reduced lead set 3 lead sets (I, II, V2)—4 electrodes Universal transformation matrix
Finlay [13]	N=744 (normal + MI + LVHs)	0.907 (0.867, 0.933)	MLR	PCA Eigenleads with BSPM 3 vectors—6 electrodes Universal transformation matrix
Tomašić [23]	N=40 (normal + patients)	-	MLR	35 channel ECG 3 vectors—4 electrodes Universal position Personalized transformation matrix median CC for lead III: 0.84
Our work	N=14 (normal)	0.920 (0.855, 0.943)	ANN	35 channel ECG 3 vectors—4 electrodes median CC for lead III: 0.87

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
