# Peer review of "Optimal Lead Position in Patch-Type Monitoring Sensors for Reconstructing 12-Lead ECG Signals with Universal Transformation Coefficient"

_sensors, 2020, doi:10.3390/s20040963_

Round 1

Reviewer 1 Report

The author proposes a method that for reconstructing a 12-lead electrocardiograph (ECG) with a universal transformation coefficient and finds the appropriate electrode position and shape. The paper need to revise based on the following comments and can be accepted after the minor comments are justified.

In the abstract, line 20, three electrode shape should be listed. In line 53 on page 2, EASI only is an abbreviation, what is the EASI system? you need to explain it clearly. In line 235 on page 7, the author said that the highest mean CC was 0.948 among all combinations in the ANN. But it is inconsistent with the data(0.954) in Table 3. Please explain it. Please increase the quality of the all figures to make the words clearly, especially for the figure 3. In line 179 on page 5, this formula should be consistent with others, the comma at the end is unnecessary. In line 131, WCT should be marked as Wilson’s central terminal. The following papers may be of great helpful to you and I think the authors could refer some papers. For example:

[1] Population based ant colony optimization for reconstructing ECG signals. Evolutionary Intelligence,2016,9(3): 55-66.

[2]Genetic algorithm for the optimization of features and neural networks in ECG signals classification. Scientific Reports, 2017, 7: 41011.

[3] Reconstruction of the 12-lead ECG using a novel MR-compatible ECG sensor network. Magnetic resonance in medicine,2019,82(5): 1929-1945.

[4] A novel method based on convolutional neural networks for deriving standard 12-lead ECG from serial 3-lead ECG. Frontiers of Information Technology & Electronic Engineering,2019,20(3):405-413.

Author Response

Thank you for your thorough review and useful comments. Please see the attachment.

Reviewer 2 Report

Abstract

On, “An artificial neural network was applied to train the transformation coefficient.”

Please specify the specification of the ANN used in this study.

On, “The mean correlation coefficient (CC) and minimum CC methods were applied to evaluate the  reconstruction performance.”

What are the values?

Introduction,

It is not clear what is the innovation of the authors compared with the-state-of-the-art. What is the gap in the studies in the literature?

2.1 Participants

Please indicate that the survey is along with the Declaration of Helsinki.

2.2 Experimental protocol

On “An ECG monitoring device that can measure 35-channel chest ECG signals was developed with a commercial ECG sensor module (PSL-iECG, PhysioLab, Korea).”

Did the system used in the study pass national/international regulations with traditional standards such as the CE mark?

Is the system used in this study, battery-powered? Please specify.

Please provide the bandwidth and the overall gain of the amplifier and also the input impedance.

The authors used the sampling rate of 250 Hz. However, based on the AHA recommendations, 500 Hz is the minimum acceptable sampling rate:Hz

https://www.ahajournals.org/doi/pdf/10.1161/01.CIR.81.2.730

Why was this sampling rate used? Please specify in the paper; this sampling rate is suitable for such applications.

On “2.3.1. Data preparation”

The digital filter of 0.5–35 Hz was used by the authors. It is related to monitoring ECG rather than diagnostics ECG. This is a significant limitation of the method that was mentioned in the conclusion of the paper. However, this is a substantial limitation of the study.

EQ 2,3 and 4: Please provide a reference for these formulas.

Fig.2: please provide x and y-axis units.

The validation procedure,

The authors used the hold-out validation. Cross-validation is needed to guard against Type III error.

The authors used a t-test for validation. Did the authors check the normality of the data?

Results:

Table 3, R-square, and ICC must be provided as well-known regression indices. CC is not a robust measure.

The Bland-Altman is needed to analyze the quality of the estimation procedure:

https://www.ncbi.nlm.nih.gov/pmc/articles/PMC4470095/

Fig.5, please improve the quality of the plot.

Tables 4 and 5, please add R-square.

Author Response

Thank you for your thorough review and useful comments. Please find our point-by point responses in the attachment.

Round 2

Reviewer 2 Report

The authors made the required modifications in the revised manuscript. However, the following two issues, as addressed by the reviewer in the previous revision, were not completely taken into account. They must be considered in the current revision so that the paper is accepted for publication:

The Bland-Altman plot must be elaborated in terms of exceeding the upper and lower acceptable limits. It must be explicitly mentioned that the method is not suitable for diagnosis based on the question of the reviewer and the answer of the authors. In the previous revision, the authors replied to the reviewer, but proper CLEAR statements were not added to the revised paper. Moreover, "monitoring" must be added to the title of the article. 

" Response 9: Thank you for your question. We could not change the cutoff frequency of highpass filter, because the device that we used has a bandwidth of cutoff frequency from 0.3 to 35 Hz as it was designed as an ECG monitoring module. As mentioned above, the aim of this paper is to examine the possibility of the universal electrode position and shape for 12-lead ECG reconstruction, not to make a device for diagnose cardiovascular diseases. In further study, we will improve our measurement device by using a lower cutoff frequency such as 0.05 Hz, and apply other methods to remove the baseline wandering "

Author Response

(The authors gave the same response as above.)
